# Overseas reception of English translations of *Journey to the West*: Temporal dynamics, cross-platform sentiment patterns, and topic modeling

**Ningning Jia**◉*, **Jing Xin, Yan Wang**

School of Foreign Languages, Wuhan Business University, Wuhan, Hubei, People's Republic of China

* nnjia_ac@163.com

## Abstract

The overseas reception of classical literature through online platforms presents a critical lens for understanding cross-cultural dynamics in the digital age. This study investigates the overseas reception of English translations of *Journey to the West* by analyzing a corpus of 1,795 reviews from Amazon and Goodreads to examine temporal dynamics, cross-platform sentiment patterns, and topic modeling. The analysis covers four celebrated translators: Arthur Waley, Anthony C. Yu, Julia Lovell, and W.J.F. Jenner. Methodologically, we developed a hybrid sentiment lexicon by integrating a domain sentiment lexicon with AFINN, NRC, and VADER through weighted fusion, addressing the limited adaptability of general sentiment lexicons in translated literature analysis. LDA modeling was further applied to enable data-driven theme extraction. Key findings reveal a consistent year-on-year increase in review counts across all translations. Notably, despite an overall positive sentiment, significant cross-platform divergences emerge, reflecting the distinct evaluative mechanisms of digital platforms. Thematic analysis identifies three central reader concerns: translation quality, plot acceptance, and character portrayal, with plot acceptance exhibiting markedly higher negativity. Furthermore, translator-level analysis reveals performance variations across these themes. This study demonstrates how digital platforms reconfigure the valuation of literary translation, and pioneers a methodological framework for capturing the dynamic interplay between reader perception, media infrastructure, and textual mobility, offering new pathways for digital humanities research in translation studies.

## Introduction

*Journey to the West* (Xiyouji, 西遊記; hereafter JTTW), a 16th-century Chinese literary classic attributed to Wu Cheng'en, tells the story of the Buddhist monk Tang Sanzang and his three disciples, Monkey King Sun Wukong, Zhu Bajie, and Sha Wujing on their journey to retrieve sacred Buddhist sutras from India [1]. Its allegorical

**Data availability statement:** The reviews from Goodreads and Amazon cannot be publicly shared because of the respective platforms' Terms. The analytical results generated in this study are provided in the Supporting Information files. The codes and domain sentiment lexicon are available at GitHub (https://github.com/Dandanjun518/JTTW_ Overseas_Reception_Analysis) and archived on Zenodo (DOI: https://doi.org/10.5281/zeno-do.17735145). For further inquiries, interested researchers may contact the corresponding author.

**Funding:** The author(s) received no specific funding for this work.

**Competing interests:** The authors have declared that no competing interests exist.

narrative, rooted in historical events but enriched with fantasy, explores universal themes of rebellion, redemption, and spiritual growth through this iconic trio's cooperation and rivalry [2]. As one of China's Four Great Classical Novels and a cornerstone of East Asian cultural heritage [3], it has achieved worldwide acclaim through its translations into multiple languages, fostering its global dissemination and cultural resonance [4]. Its notable English-language translations include Arthur Waley's *Monkey* (abridged), Julia Lovell's *Monkey King: Journey to the West* (abridged), and the unabridged versions by Anthony C. Yu (*The Journey to the West*) and W. J. F. Jenner (*Journey to the West*).

Studying the overseas reception of translated literary works is vital for assessing their cross-cultural communicative impact and efficacy in cultural dissemination. Within this evaluative framework, reader reception serves as a key metric for evaluating the effectiveness of translated works in cross-cultural contexts [5]. The significance of reader response is further underscored by multiple theoretical perspectives. Nida's Functional Equivalence theory posits that translation effectiveness fundamentally hinges on achieving an equivalent reaction between target-language readers and source-language readers [6]. This concept aligns closely with Reader-Response Criticism, which conceptualizes reception as a dynamic process of real-time text reconstruction within an aesthetic temporal framework [7], highlighting readers' active role in constructing meaning rather than passively absorbing fixed interpretations. Building on these theoretical foundations, participant-oriented translation studies further emphasize the critical necessity of engaging with actual readers' feedback, particularly within contemporary digital contexts [8].

Sentiment analysis (SA), the computational detection of emotional tones in text, has gained prominence in literary translation research as a tool to analyze readers' opinions, sentiments, appraisals, attitudes, and emotions [9]. The application of SA in translated literature represents a burgeoning field, with scholars increasingly using it to interrogate sentiment polarity in cross-lingual contexts [10]. Current SA methodologies primarily fall into two categories: deep learning models and lexicon-based approaches. Deep learning models, particularly transformer-based models like BERT and large language models (LLMs), face significant hurdles when applied to niche domains such as reviews of classical literary translations. A primary constraint is their data-hungry nature. BERT requires substantial data to perform reliably [11–12], a requirement that is equally, if not more, pronounced for LLMs [13–14]. This poses a particular challenge for specialized studies where sample sizes are often limited to a few thousand reviews or fewer. Compounding this issue is a domain adaptation gap. As these models are pre-trained on general corpora, they often exhibit weak recognition of nuanced, culture-specific expressions [15]. Furthermore, each model family has distinct shortcomings. For instance, the context window of BERT can hinder its ability to capture emotional correlations across long reviews [16], while LLMs are particularly prone to generating factually incorrect or hallucinated emotional interpretations [17]. Given these constraints, lexicon-based methods offer a compelling alternative due to their interpretability, lower computational cost, and efficacy with small datasets.

However, prevailing sentiment lexicons crystallize two persistent challenges for analyzing literary translation reviews. First, general sentiment lexicons frequently overlook domain-specific vocabulary [18]. Specifically, when readers label a translation "literal", they implicitly condemn the translator for failing to enact cultural adaptation [19], yet general sentiment lexicons neglect the disciplinary semantic shift inherent in such criticisms, thereby failing to recognize this negative valuation. Secondly, general sentiment lexicons are prone to misclassifying sentiment-bearing words within specific literary contexts that diverge from their conventional lexical classifications [20]. To be specific, terms like "rebellious" are usually categorized as negative based on general usage patterns. However, in reviews of JTTW, descriptions of Sun Wukong as "rebellious" overwhelmingly convey admiration, signifying a distinct polarity inversion driven by cultural archetypes. This phenomenon underscores the significant impact of cultural context and narrative framing on sentiment interpretation, necessitating domain-adaptive approaches or specialized modeling to accurately capture sentiment in culturally rich narratives [21]. Consequently, building tailored sentiment lexicons emerges as a critical imperative to capture context-dependent nuances overlooked by general SA tools [22–23].

In recent years, the thematic analysis of reader reviews in translated literature has become a critical area of cross-cultural research. Current studies on thematic analysis of reader reviews in translated literature largely depend on manual qualitative interpretation [24–25], since most reviews are sparse (often around one hundred to several hundred). This approach involves researchers manually examining comment content to identify recurring themes through close reading. Apart from this, a few scholars attempt to employ computational tools to mine themes, such as NVivo [8] and BERTopic [26]. Among these computational methodologies, Latent Dirichlet Allocation (LDA) stands out as a generative probabilistic model designed to uncover abstract themes within text corpora. In contemporary practice, it has been widely applied in text mining, including tasks such as document clustering [27] and information retrieval [28]. Significantly, LDA has gained significant traction in literary studies for its ability to uncover thematic structures and facilitate genre-based clustering across diverse literary corpora, such as children's literature [29] and novels [30]. As LDA has flourished in literary research, scholars have increasingly extended its use beyond traditional literary genres to analyze thematic structures in reader reviews, applying topic modeling to uncover patterns of audience interpretation and critical discourse. This shift reflects a broader trend toward integrating computational methods to explore both textual content and reader engagement in literary studies. Building on established methodologies for thematic mining of reader reviews [31], this study extends LDA application to a more expansive corpus, seeking to uncover nuanced patterns of audience perception and interpretive trends in cross-cultural literary reception.

Given their capabilities, this study employs these computational methods to quantitatively investigate the core concerns of the theoretical frameworks outlined above. Specifically, sentiment analysis operationalizes the core concept of Nida's Functional Equivalence theory by quantifying the valence and intensity of readers' affective responses. This provides an empirical basis for evaluating the nature of the target-language readers' emotional response, a key factor in assessing a translation's functional equivalence. Simultaneously, LDA topic modeling offers a means to capture the dynamic, constructive nature of reception emphasized by Reader-Response Criticism. By inductively identifying the emergent themes that preoccupy readers, LDA reveals the collective hermeneutic processes through which meaning is negotiated and assigned to the translated work. Thus, the integration of sentiment analysis and topic modeling facilitates a large-scale, empirical inquiry into the reception process, effectively bridging theoretical postulation with data-driven analysis of how translated literature is actually received and interpreted by a global readership.

Based on the foregoing review, this study examines the overseas reception of the English translations of JTTW. It is crucial to clarify that the term "overseas reception" herein does not denote its broadest sense. Rather, it is operationalized specifically as the English-language overseas online reception manifested in reviews on two major global platforms of Amazon and Goodreads. This delineated, Anglophone, platform-based focus enables a detailed, data-driven analysis. Accordingly, the study addresses the following three research questions.

(1) How do longitudinal trends in review counts evolve for four key English translations of JTTW, both collectively and for individual editions?

(2) What comparative sentiment patterns emerge between Amazon and Goodreads for JTTW translations, and how do these patterns vary across the translators?

(3) What thematic clusters structure reader reviews of English translations of JTTW, and how do sentiment patterns vary within each cluster and across translators?

Collectively, these inquiries seek to unpack the multifaceted overseas reception of JTTW translations. By integrating interdisciplinary methodologies—combining domain sentiment lexicons, computational topic modeling, and cross-platform comparative frameworks—this study endeavors to address these research questions, with subsequent analyses revealing nuanced dynamics in reader engagement, translational strategies, and the evolving reception of classical Chinese literature in digital spaces.

## Materials and methods

### Data collection and preprocessing

We selected reviews of the four most-reviewed JTTW translations—by Arthur Waley, Anthony C. Yu, Julia Lovell, and W.J.F. Jenner—as research data. This approach, preferable to single-translator analysis or unfiltered keyword searches (e.g., "Journey to the West"), addresses three fundamental constraints. First, it ensures substantial data for computationally intensive methods such as LDA topic modeling and sentiment analysis, as individual translators' works typically attract insufficient review counts. Second, it enables examination of translational styles and platform-mediated reception dynamics, core dimensions in reception studies. Third, it mitigates data noise by excluding irrelevant responses inherent in indiscriminate collection.

Having defined the corpus through translator selection, we deliberately delimited the data sources to two major online platforms, Amazon and Goodreads. This selection is justified by their distinct and complementary positions within the literary ecosystem. Goodreads, described as "the Anglophone world's dominant book-centric social networking platform" [32], specializes in community-driven discourse exploring textual nuances and reading experiences [33]. Conversely, Amazon, as a premier commercial retailer, generates user reviews focused on practical, purchase-related evaluations [34]. This strategic focus on two platform types—social-literary and commercial—ensures the analysis captures a broad spectrum of reader responses, from literary evaluation to utilitarian judgment.

A total of 2,118 reviews were collected from Amazon and Goodreads as of March 13, 2025, as documented in Table 1. Notably, the data collection from Amazon was subject to a platform-imposed sampling constraint. Following a post-2023 interface update, Amazon product pages algorithmically curate and display only the top 100 "most relevant" reviews to users. This proprietary ranking system determines relevance by synthesizing multiple factors, including, but not limited to, review recency, whether the purchase was verified, and the level of user engagement [35]. Consequently, the obtained Amazon corpus is not a random or chronologically sequential sample; it is a platform-filtered subset that reflects Amazon's internal ranking mechanisms.

**Table 1. Edition profiles and cross-platform review counts of four English translations of JTTW.**

| Translator | Edition title | Publication year | Amazon reviews | Goodreads reviews | Total |
|---|---|---|---|---|---|
| Arthur Waley | *Monkey: The Journey to the West* | 1994 | 100 | 717 | 817 |
| Anthony C. Yu | *The Journey to the West* (Volume 1–4) | 1980 & 2012 (Revised) | 186 | 307 | 493 |
| Julia Lovell | *Monkey King: Journey to the West* | 2021 | 64 | 419 | 483 |
| W.J.F. Jenner | *Journey to the West* | 1993 | 100 | 225 | 325 |

                                                                

This constraint directly defined the upper limit of retrievable reviews for each book sourced from Amazon. A notable exception to this per-title limit arises from the handling of multi-volume works. Yu's four-volume translation is cataloged as separate editions on Amazon, with each volume maintaining an independent review pool. This bibliographic segregation explains the cumulative total of nearly 200 Amazon reviews across the series. However, on Goodreads, the cataloging system combines reviews of Yu's translation for both the inaugural 1980 edition and its 2012 revised iteration into a single bibliographic entry.

Our data collection and analysis were conducted in compliance with Amazon's and Goodreads' terms of service, which permit non-commercial academic study. The reviews were collected via Python-based web scraping from publicly accessible pages, consistent with established practices in computational literary studies [36]. To protect contributor privacy, all reviews were anonymized upon collection, in accordance with principles outlined in ethical guidelines for internet research [37]. Given that the study involved only the analysis of existing, publicly available, and anonymized data, it was not considered human subjects research requiring ethical approval. The collected reviews were processed through a sequential preprocessing pipeline to ensure data quality and analytical robustness. The initial phase involved raw text cleaning, including removal of HTML tags, URLs, and non-alphanumeric symbols, followed by whitespace normalization and case unification. Subsequently, all non-English entries along with duplicate and non-substantive comments were systematically excluded, yielding a corpus of validated 1,795 English reviews. Furthermore, to ensure robust thematic and sentiment analysis, we then performed key conceptual normalization steps. Multi-word proper nouns and domain-specific key terms (e.g., "Sun Wukong") were merged into single tokens using underscores (e.g., "sun_wukong"). Crucially, variant translations referring to the same entity across different editions were mapped to a canonical form. For instance, all mentions of the character Sha Wujing, including "Sandy" from the Waley translation and "Friar Sand" from the Jenner translation, were normalized to the single token "sha_wujing". This process guarantees that subsequent analyses are based on coherent lexical units, ensuring accurate thematic clustering and sentiment measurement.

## Sentiment analysis

To address the challenge of sentiment analysis in translated literature reception research, we propose a hybrid approach (Fig 1) that integrates a domain sentiment lexicon with three well-established general sentiment lexicons (AFINN, NRC and VADER) through a weighted mechanism.

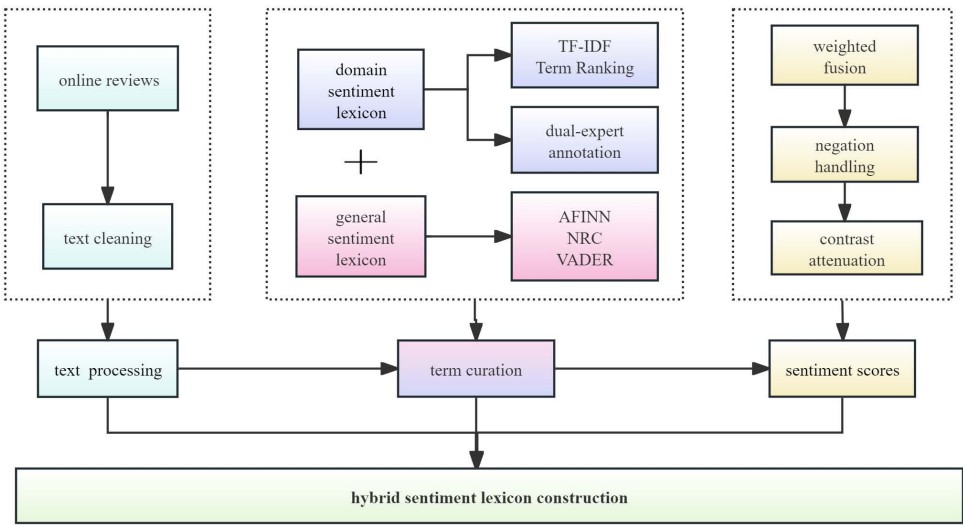

**Fig 1. Hybrid sentiment lexicon construction pipeline.**

Primarily, we constructed the domain sentiment lexicon through a multi-phase protocol. First, high-frequency terms were extracted from processed review data using TF-IDF weighting. The two annotators (both associate professors in translation studies) first collaborated to develop annotation guidelines, establishing shared criteria for the 7-point polarity scale (−3 to +3). This granularity level, selected as optimal for sentiment lexicon development, balances discrimination capacity and rater reliability, a methodological calibration empirically validated in human rating behavior studies [38]. Following this preparatory phase, they independently annotated the extracted terms. All cases of disagreement were subsequently resolved through a consensus discussion between the two annotators, resulting in a single, agreed-upon score for each term. Inter-annotator agreement, measured by Cohen's Kappa, was 0.77, indicating substantial agreement. The resultant domain lexicon comprises 448 calibrated entries, encompassing high-frequency translation terms and sentiment-bearing expressions unique to cross-cultural literary interpretation. Within the sentiment analysis pipeline, this domain lexicon serves as the primary resource; terms not captured therein are addressed by a weighted fusion of general sentiment lexicons.

Three general sentiment lexicons are used to complement the domain lexicon, namely NRC, AFINN, and VADER. The NRC Emotion Lexicon excels at capturing discrete, fine-grained emotions [39], making it well-suited for deciphering complex emotional expressions in literary reviews. Similarly, AFINN is distinguished by its high reliability in sentiment polarity judgment and ability to quantify emotional intensity [40], which aligns with the need to assess the strength of readers' responses. VADER, optimized for informal short texts, demonstrates robustness in processing concise reviews [41], complementing the others in capturing context-sensitive cues. Rather than applying equal weights, which would overlook the lexicons' heterogeneous contributions [42], we adopted a weighted fusion scheme guided by the established methodological principle that integrating multiple sentiment lexicons should be based on their adaptability to the target domain and their contribution to overall performance [43–45], an approach validated in prior lexicon construction studies [46]. Following this principle, NRC and AFINN were assigned higher weights due to their higher inherent adaptability to the stylistic and evaluative nature of literary reviews. Conversely, VADER, while valuable for informal language, received a lower weight owing to its primary optimization for social media contexts, which presents potential domain adaptation limitations for longer-form literary reviews. This theoretically informed weighting preference was empirically tested through the grid-search optimization described after the full pipeline is outlined.

The pipeline also incorporates a series of contextual and normalization adjustments, including handling negation, contrastive discourse, and scale normalization, to ensure that the final sentiment scores reflect nuanced contextual cues beyond mere lexical choice. To model negation effects, we adopt a 2-token backward window to count negators (e.g., "not", "never") within the contextual valence shifter framework. This approach systematically reverses sentiment polarity through syntactic scope determination, a method empirically validated for negation scope detection in review classification [47]. Subsequently, for contrastive discourse markers such as "but" and "however", we implemented a distance-based weight attenuation mechanism to adjust emphasis decay within discourse structures [48]. The scores, after these adjustments, are aggregated and then normalized via a hyperbolic tangent transformation to the range [− 1, 1], ensuring scale consistency across heterogeneous sources, a critical step for robust fusion modeling [49]. This yields a final, continuous sentiment score for each review, which is then discretized into positive, negative, and neutral categories via threshold-based classification.

Thus, to determine the optimal lexicon weights and classification threshold for this pipeline described above, we conducted a systematic grid-search optimization based on a manually annotated tuning set. The tuning set consisted of 180 comments (approximately 10% of the total corpus), randomly sampled with attention to comment length and translator representation, and independently annotated by two annotators (both associate professors). The annotators judged the overall sentiment polarity (positive, negative, or neutral) of each comment based on its dominant emotional tone; disagreements were resolved through discussion. Inter-annotator agreement reached a substantial level (Cohen's Kappa = 0.82), confirming the reliability of this benchmark dataset. A grid search was then performed on this tuning set to

optimize the lexicon weights and sentiment threshold. The search explored weight combinations for AFINN (range: 0.3 to 0.5), NRC (0.3 to 0.5), and VADER (0.1 to 0.3) with a step size of 0.1, as well as threshold values of 0.05, 0.1, and 0.15. Macro-F1 was selected as the optimization criterion because it weights all classes equally, providing a performance assessment that is robust to class imbalance [50], a crucial characteristic given the predominance of positive reviews in our corpus. The configuration yielding the highest macro-F1 score (79.45%) on the tuning set was a weight set of 0.4 (AFINN), 0.4 (NRC), and 0.2 (VADER) with a threshold of ±0.1; this was adopted as the final model. These empirically optimized parameters define the final scoring and classification rules, formalized below.

$$s(w) = \begin{cases} \text{score}_{\mathcal{D}_c}(w) & \text{if } w \in \mathcal{D}_c \\ 0.4 \cdot \text{NRC}(w) + 0.4 \cdot \text{AFINN}(w) + 0.2 \cdot \text{VADER}(w) & \text{otherwise} \end{cases}$$

$$\text{class}(T) = \begin{cases} \text{Positive} & \hat{S} > 0.1 \\ \text{Negative} & \hat{S} < -0.1 \\ \text{Neutral} & \text{otherwise} \end{cases}$$

Lastly, to verify the robustness of the analytical framework, we applied minor perturbations to the weights (±0.05, half the grid step size) and threshold (±0.02) and re-evaluated the sentiment analysis. The key empirical findings and statistically significant patterns remained stable across all parameter variations, confirming that the core conclusions are not artifacts of a specific parameter setting. Together, the grid-search optimization and the sensitivity analysis ensure that the analytical framework is both data-driven and robust, providing a reliable foundation for the subsequent analysis.

## LDA topic number selection

We determined the optimal number of topics using an LDA model through a dual validation framework combining statistical metrics and domain interpretability. Following established practices in topic modeling, we first computed perplexity scores across candidate topic numbers. The result showed that the optimal value occurred at K = 4 (−7.241), closely followed by K = 3 (−7.242). Although K = 4 achieved marginally better perplexity, manual inspection of topic-word distributions reveals that K = 3 generated three semantically cohesive themes highly relevant to translated literature analysis, whereas K = 4 introduced over-fragmented topics with limited practical relevance.

To further verify this domain-based insight and ensure the selected topic number K = 3 also exhibited strong semantic coherence, we conducted a coherence score calculation using the *Cv* metric. The topic coherence evaluation revealed that K = 3 demonstrated the second-highest score at 0.4356. Notably, this coherence score for K = 3 exceeded that of K = 4 (0.4280), suggesting better topic stability for three-topic modeling. Integrating both perplexity and coherence, we finalized K = 3 as the optimal choice. This dual-metric approach for resolving perplexity ambiguities ensures alignment with both statistical rigor [51] and domain-specific interpretability—essential for translation analysis where topic meaningfulness directly impacts downstream tasks [52]. It is noteworthy that while K = 3 was determined to be optimal for identifying the high-level thematic structure central to our research questions, it should be acknowledged that a slightly higher K (e.g., K = 4) still has the potential to uncover more nuanced subthemes.

After determining the optimal topic number through combined perplexity-coherence assessment, we performed topic-level sentiment analysis using the hybrid lexicon approach proposed earlier. Following empirical validation of accuracy improvement through hard assignment [53], each review was assigned to its single highest-probability topic derived from LDA distributions. This document-topic mapping enabled granular emotional analysis by applying the lexicon to texts grouped by dominant topic.

## Results and discussion

### Temporal dynamics of review counts: Holistic and translator-specific

The temporal evolution of reader reviews for English translations of JTTW is analyzed through complementary lenses: a macro-level perspective delineates the cumulative growth trajectory from 1999 to 2024 (Fig 2), with data from January to March 2025 excluded due to incomplete annual representation; concurrently, a translator-centric perspective contrasts distinct growth patterns across the Waley, Yu, Lovell, and Jenner editions (Fig 3).

As is shown in Fig 2, the results demonstrate a sustained upward trajectory in annual review counts from under 50 reviews per year before 2012 to nearly 300 by 2024, with particularly accelerated growth post-2013 and a striking surge during 2021–2024. This overall growth pattern suggests a plausible association with a rising global interest in the English

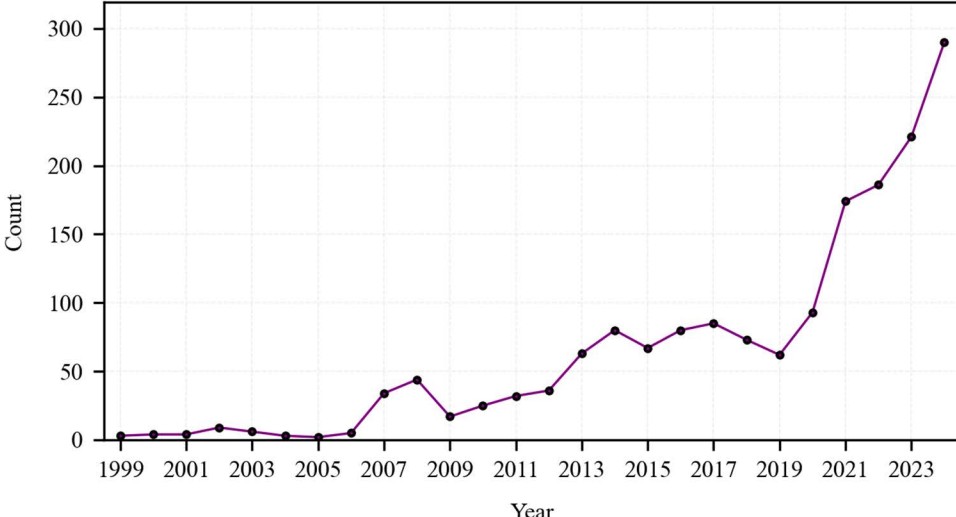

**Fig 2. Longitudinal trend of annual review counts (1999-2024).**

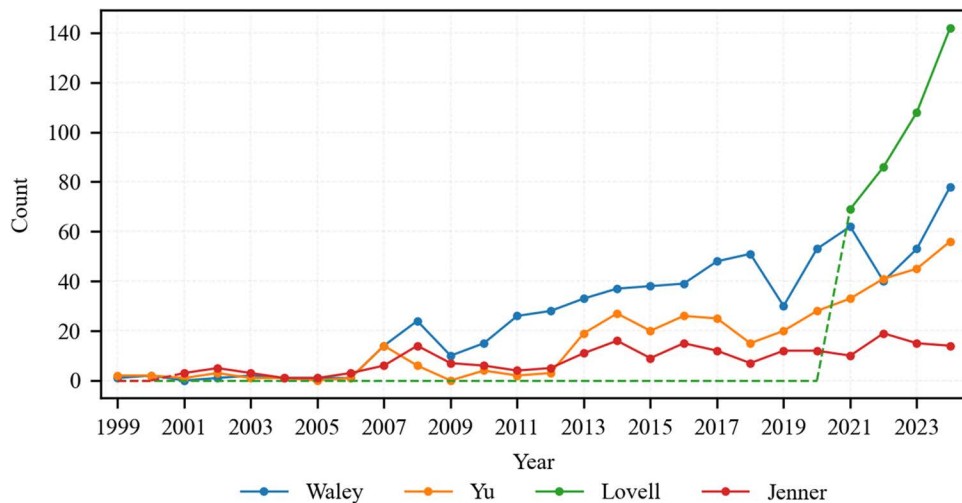

**Fig 3. Longitudinal trend of annual review counts for individual translators (1999-2024).**

translations of JTTW, potentially amplified by a series of influential multimedia adaptations. The 2015 animated film *Monkey King: Hero Is Back* has garnered acclaim for its innovative character design and dynamic storytelling [54]. Its international success seems to correlate with an early growth phase in the reception of the novel's English translations, as suggested by reviews that directly reference the film: "*Monkey King* definitely is a fantastically funny paver of pilgrimage tales. It creates a character fixed in fable and folktale, and is the subject of treatments from short story to book, from oral legend to the animated character onscreen (*Monkey King: Hero is Back*)." Subsequently, the 2019 blockbuster *Ne Zha* achieved global box office success by leveraging its subversive retelling of mythological themes, sparking broader interest in the *Journey to the West* universe [55]. This widespread cultural resonance may have further sustained audience engagement with the literary work. The 2021 release of the English-dubbed CCTV version of *Journey to the West* (1986 TV Series), which gained unexpected traction on platforms like YouTube and bilibili [56], could have contributed to this continuity, as evidenced by reviews explicitly recommending it: "…instead encourage you to learn to read the Chinese original…one of the many movies/TV productions available is a great help in enjoying this work. I highly recommend the CCTV production from mainland China…" Most significantly, the steep surge observed in 2024 shows a strong temporal correlation with the launch of the video game *Black Myth: Wukong*. Overseas players, drawn to the game's immersive reimagining of mythological landscapes and combat mechanics, have increasingly sought out English translations of the novel to more deeply decode in-game narratives [57]. This correlation is also substantiated by multiple reviews that credit the game for prompting their engagement, such as one stating, "The video game *Black Myth: WuKong* came out recently and looked amazing which brought *Journey to the West* and its many adaptations to my attention," and another explaining, "I read this after playing *Black Myth: Wukong*... and needed to understand the story behind it."

What is exhibited in Fig 3 uncovers distinctive growth patterns among the four translators: Waley's late-stage resurgence peaking near 80 reviews per year, Yu's steady upward progression approaching 60 reviews, Lovell's dramatic emergence with a leap to 140 reviews following its debut in 2021, and Jenner's consistent reception intensity below 20 reviews. This variation is explicated through the power dynamics of patronage systems and cultural capital accumulation, as theorized in Lefevere's patronage and Bourdieu's framework.

Primarily, Lefevere's patronage theory posits that institutional patronage fundamentally governs translation dissemination [58]. Waley's *Monkey* and Lovell's translation, both published by Penguin Classics, have attained the status of classics in Western Sinology. Penguin Classics is renowned for its global distribution and the high regard it commands in the literary world. It has been pivotal in introducing numerous world literary masterpieces to a wide readership [4]. Its reputation for quality and wide reach, as seen in its history of making literature accessible to the masses [59], has contributed to the strong reception of Waley's and Lovell's works. Concurrently, Yu's four-volume translation, published by the University of Chicago Press, benefits from that publisher's rigorous academic reputation and scholarly community integration [60]. Conversely, Jenner's translation is published by Foreign Languages Press. While this press has been involved in promoting Chinese literature overseas, compared to Penguin Classics and the University of Chicago Press, it may have a more limited reach in the Western market. It lacks significant mainstream channel marketing, and its circulation is mainly confined to academic libraries. This oversight risk aligns with established scholarship on patronage deficits, where insufficient institutional support in marketing and distribution precipitates translational invisibility [61].

Complementarily, translators' accrued cultural capital constitutes a pivotal factor. Arthur Waley's stature as a foundational Western Sinologist and his groundbreaking translation established enduring readership engagement. Anthony C. Yu's University of Chicago professorship and American Academy of Arts and Sciences membership conferred symbolic capital that authenticated his work academically. Julia Lovell's position at University College London, coupled with her prolific contributions to elite media (*The Guardian, Financial Times, The New York Times*), cultivated transnational reader trust. Her translation's visibility was further amplified by celebrity endorsements, notably Gene Luen Yang's preface and the *Los Angeles Review of Books* feature. By contrast, W.J.F. Jenner's institutional confinement to the Foreign Languages Press reflects cultural capital limitations within Western literary spheres.

 

## Cross-platform sentiment patterns and translator variance

Generally speaking, sentiment analysis is conducted in the following three phases. First, using our hybrid sentiment lexicon methodology, we conducted an overall sentiment analysis of reader reviews. Results (Fig 4) indicate a pronounced dominance of positive sentiment, with precisely 83.84% of reviews classified as positive, contrasting sharply with 8.19% negative and 7.97% neutral sentiment. This distribution underscores an overwhelmingly favorable reception of the English translations within overseas readerships.

Subsequently, following the overall sentiment analysis, we conducted a further comparative analysis between Amazon and Goodreads, which revealed significant distributional divergences across platforms. As is shown in Fig 5, Amazon displays higher positive proportions and lower negative sentiment compared to Goodreads. Particularly noteworthy is the widening gap during the 2015−2025 period, suggesting increasingly divergent reception patterns. To quantitatively validate these observed differences, we conducted statistical comparisons as shown in Table 2. Amazon exhibited a significantly higher positive sentiment proportion than Goodreads (88.73% vs. 82.41%; Δ = +6.32 pp), while showing reduced negative sentiment (5.39% vs. 9.01%, Δ = −3.62 pp). Although neutral sentiment differed moderately across platforms (Δ = −2.70 pp), a Pearson's chi-square test confirmed the overall sentiment distributions varied significantly between platforms ($\chi^2 = 9.42$, df = 2, $p = 0.009$).

These divergent review patterns between Amazon and Goodreads can be traced to their distinct core functionalities, which fundamentally shape their respective user bases and, consequently, the evaluation mechanisms that emerge on each platform. Amazon's framework is intrinsically built around its e-commerce model. This commercial orientation prioritizes purchase-related contexts, which in turn attracts a broad and diverse user base that includes many casual buyers [62]. These users often approach book reviews through the lens of product satisfaction, leading to evaluations based on non-literary criteria. This tendency is reflected in the frequent use of transactional keywords like "buy," "purchased," and "price" within reviews, and often results in shorter, more utility-driven comments with limited focus on literary analysis. The platform's rating system, reinforcing this mindset, encourages more extreme ratings rather than nuanced assessments of textual merit [63].

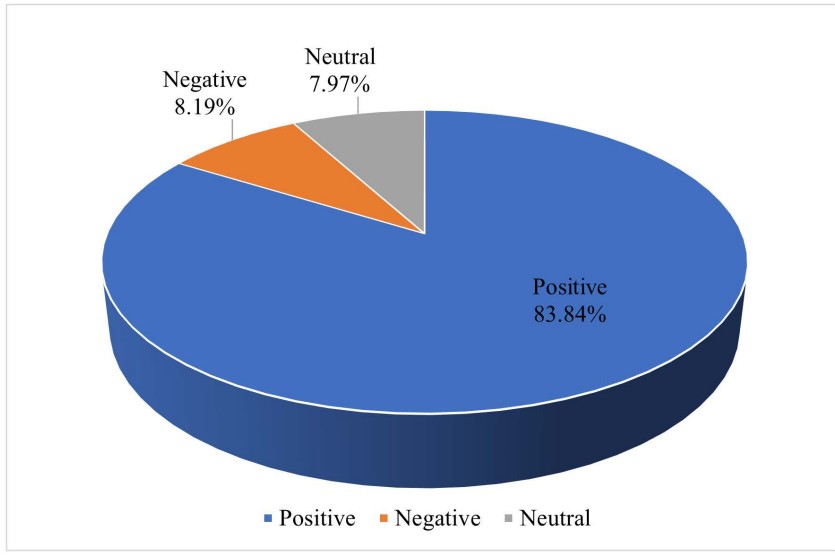

**Fig 4. Overall sentiment distribution of reviews.**

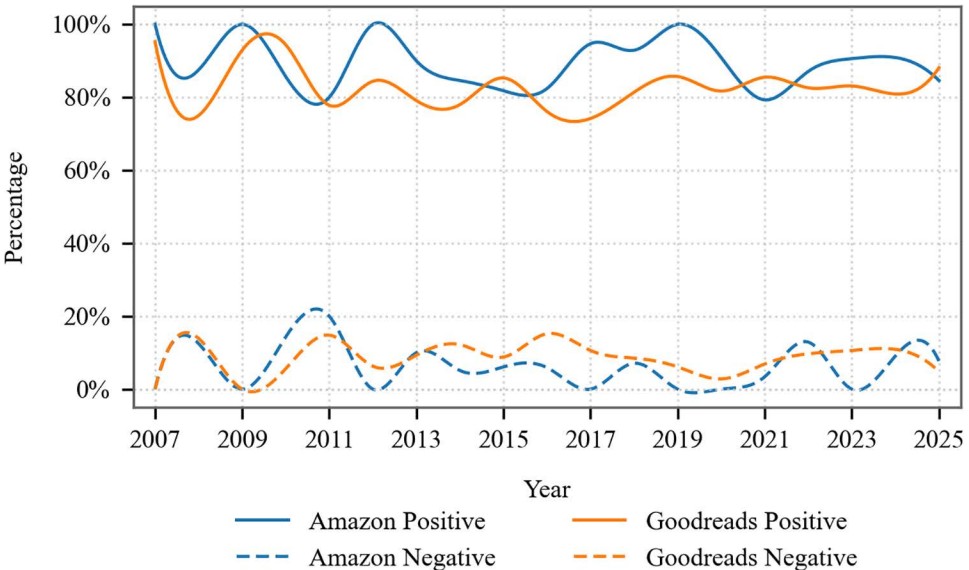

**Fig 5. Longitudinal positive and negative sentiment patterns across Amazon and Goodreads (2007–2025).** The analysis period begins in 2007, as this marks the earliest year for which publicly accessible review data exists on Goodreads.

**Table 2. Cross-platform sentiment distributions and statistical significance.**

| Sentiment | Amazon | Goodreads | Δ (pp) | *p*-value |
|---|---|---|---|---|
| Positive | 88.73% | 82.41% | +6.32 pp | *p* = 0.009 |
| Negative | 5.39% | 9.01% | −3.62 pp | |
| Neutral | 5.88% | 8.58% | −2.70 pp | |

Note: Δ (pp) = Amazon proportion – Goodreads proportion. Proportion differences were assessed using a Pearson's chi-square test. The test evaluates whether sentiment distributions as a whole differ between platforms, yielding a significant association ($\chi^2$ = 9.42, df = 2, *p* = 0.009).

In contrast, Goodreads is designed as a social-literary ecosystem centered on reading as a cultural practice [64]. This core functionality fosters a community of self-identified, engaged readers who interact with books as cultural artifacts. This user base is more likely to participate in genre discussions, update reading statuses, and use the platform's tagging system and community-driven "classics" categorization [65]. Within this environment, ratings and reviews themselves become a form of cultural capital; users employ them to perform literary expertise and engage in communal discourse [66]. Consequently, this performative aspect of reviewing inherently cultivates more nuanced, moderate ratings and reviews that emphasize narrative complexity, thematic depth, and writing style.

Ultimately, the algorithmic underpinnings of each platform solidify these differences: Amazon's recommendations cater to mass-market appeal and accessibility, while Goodreads' algorithms prioritize review depth and community engagement [32]. Thus, the observed patterns in sentiment and evaluation are a direct consequence of the foundational design principles governing each space, as seen in the contrast between e-commerce and social-literary platforms.

After exploring cross-platform sentiment disparities, we further investigate how individual translators' reception varies between Amazon and Goodreads. Given that the most significant cross-platform disparity manifests in positive review proportion, and considering the statistical robustness afforded by their larger sample size, we operationalize translator platform sensitivity through shifts in positivity proportion.

Results reveal systematic differentials in positive evaluations of literary translations. As quantified in Table 3, all translators demonstrated higher positive proportions on Amazon compared to Goodreads (mean Δ = +7.475 percentage points; range: +3.8 to +12.4 pp). Two translators exhibited statistically significant cross-platform discrepancies: Jenner manifested the largest effect size (Δ = +12.4 pp, *p = 0.026), while Yu showed the strongest statistical significance (Δ = +9.9 pp, **p = 0.004), with his scholarly edition achieving 91.8% positive reception on Amazon versus 81.9% on Goodreads. Contrastingly, translators Waley and Lovell showed directionally consistent but non-significant Amazon advantages (both Δ = +3.8 pp; p > 0.35).

Although both Yu and Jenner exhibit platform sensitivity, we foreground Yu's case, as his edition commands substantially higher reader engagement and wider scholarly recognition. Yu's translation, marketed as the first comprehensive scholarly edition, distinguishes itself through systematic foreignization strategies that preserve Buddhist and Taoist concepts and annotate cosmological frameworks [67]. His maximalist approach manifests in an 88-page introduction, exhaustive notes, and complete chapter fidelity to the source text. Within Amazon's commercial field where e-commerce orientation prioritizes transactional utility and perceived value, Yu's comprehensive scholarly apparatus, including extensive annotations, cultural fidelity, and academic rigor, functions as institutionalized cultural capital [68]. This aligns with consumer expectations of premium authority, catalyzing perceptions of superior worth. Conversely, in Goodreads' literary subfield, which valorizes immersive narrativity and aesthetic coherence, these identical features trigger habitus dissonance. Yu's comprehensive critical apparatus, including complete poetic renditions and dense notes, violates the field's fundamental illusio: uninterrupted narrative immersion [69]. Such interventions constitute symbolic violence against readerly expectations [70], reframing academic rigor as disruptive rather than enriching. Consequently, positivity registers lower than its counterparts.

Arthur Waley and Julia Lovell exhibit robust platform reception precisely because their translation habitus employs moderate domestication, preserving core narratives while avoiding philological density. This constitutes neutral cultural capital convertible in both fields, fulfilling Amazon's baseline readability requirements without provoking Goodreads' aesthetic sanctions [71].

## Topic modeling and sentiment variation

### 3.3.1. Thematic clustering and sentiment distribution.
We employed LDA topic modeling to decode latent thematic clusters in reviews. Methodologically, this complements sentiment analysis by revealing readers' focus in judgment formation. In the inter-topic distance map, as visualized in Fig 6, significant spatial dispersion confirms clear thematic discreteness, with each bubble representing one independent topic and its size proportional to the topic's prevalence within the corpus.

The most salient terms for each topic, as detailed in Table 4, collectively delineate the three discrete dimensions governing reader evaluation. Theme one is anchored by a cluster of proper nouns for central characters ("monkey", "tang_sanzang", "sun_wukong", "zhu_bajie") and the key mythological figure ("buddha"). Theme two is characterized

**Table 3. Cross-translator positive proportions and statistical significance.**

| Translator | Amazon | Goodreads | Δ (pp) | *p*-value |
|---|---|---|---|---|
| Jenner | 85.4% | 73.0% | +12.4 pp | p = 0.026* |
| Yu | 91.8% | 81.9% | +9.9 pp | p = 0.004** |
| Waley | 86.3% | 82.5% | +3.8 pp | p = 0.359 |
| Lovell | 89.4% | 85.6% | +3.8 pp | p = 0.479 |

Note: Positive proportions reflect the percentage of favorable reviews. Δ (pp) = Amazon proportion – Goodreads proportion. Statistical significance was assessed via two-tailed z-test (*p < 0.05, **p < 0.01).

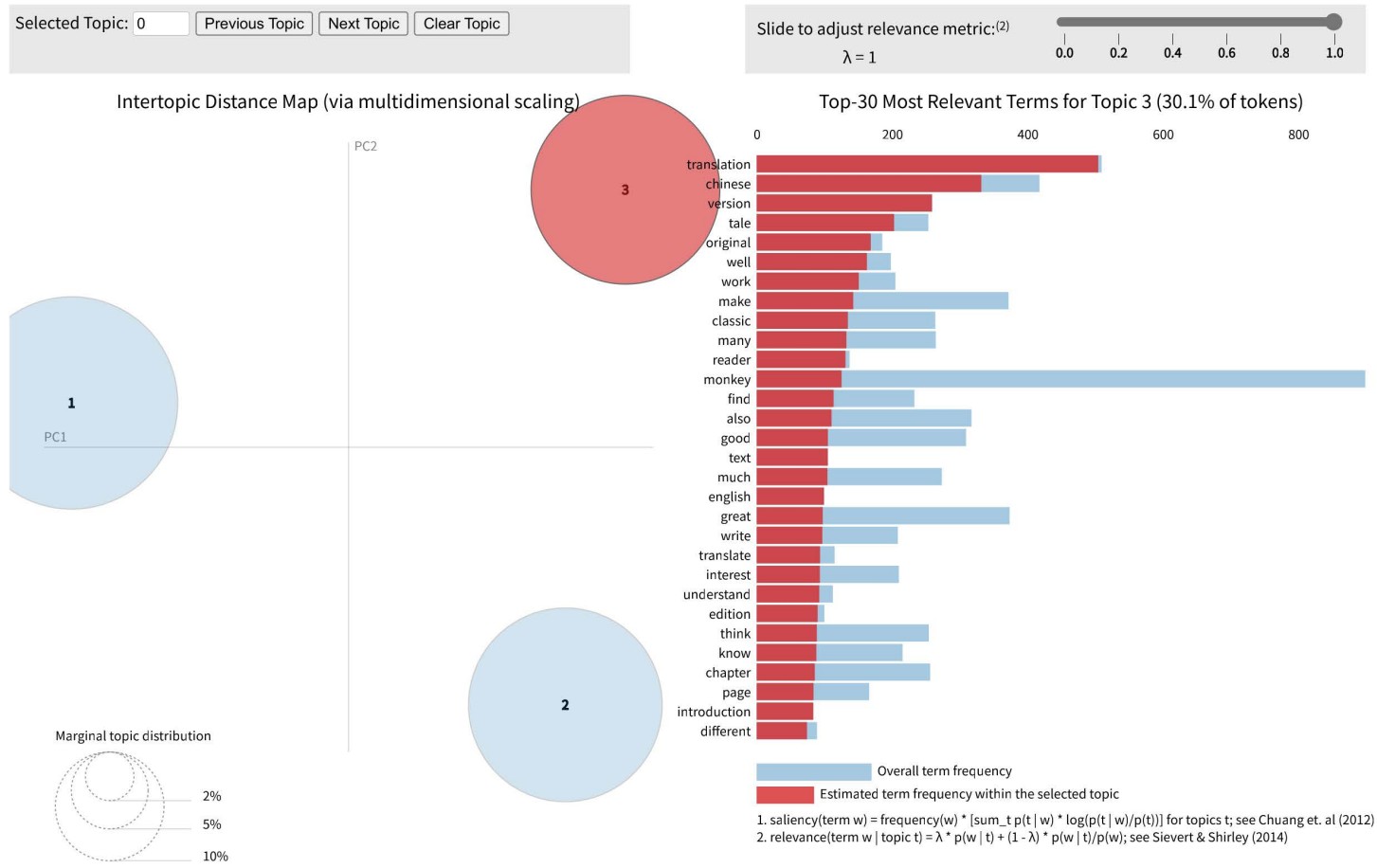

**Fig 6. LDA visualization of three themes in reviews.**

**Table 4. Themes and top ten keywords from LDA topic modeling.**

| Topic | Thematic label | Top ten keywords |
|-------|----------------|------------------|
| 1 | character portrayal | monkey tang_sanzang sun_wukong zhu_bajie heaven china demon become buddha character |
| 2 | plot acceptance | like time monkey journey really fun great good first enjoy |
| 3 | translation quality | translation chinese version tale original well work make classic many |

overwhelmingly by evaluative and experiential language ("like", "fun", "great", "good", "enjoy"), and incorporates the narrative concept of the "journey". Theme three is dominated by translation-specific lexemes ("translation", "chinese", "version", "original") and evaluative markers ("classic", "well"). These themes align with the horizon of expectation [72], reflecting readers' tripartite evaluation framework for translated literature: character portrayal, plot acceptance, and translation quality.

In translation studies, understanding the sentiment towards different themes in translated works and how such sentiment varies across translators is of great significance. To explore this further, we delved into the sentiment distributions across these themes and their variations among translators. These results, presented in Figs 7–9, will be analyzed in subsequent sections with respect to translation quality, plot acceptance, and character portrayal.

**3.3.2. Translation quality.** As evidenced in Fig 8, the semantic network reveals a densely interconnected lexical core manifesting a dual evaluative emphasis on literary reception and cultural mediation. Central descriptors "great" and "classic" signify readers' strong approval of translational merit, while their convergence with "epic" further corroborates recognition of the novel's canonical status. Simultaneously, the frequently co-occurred affect-laden terms like "recommend" reflect readers' immersive engagement with the translated text and their endorsement of its literary and cultural value. Integral to this nexus, the words "know" and "learn" further reflect readers' perceived acquisition of China-specific philosophical frameworks through the translated text. Adjacently, "culture" and "history" highlight the work's efficacy as a cross-cultural conduit. Critically, bridging terms like "enlightenment" and "understand" denote successful epistemological transfer, mediating source-text tenets with target-reader cognition.

The salient topic term "version" (Fig 8) underscores readers' engagement with translational variants. To further quantify how readers evaluate these variants, Fig 9 provides translator-specific sentiment data for translation quality: Yu's translations received the highest proportion of positive evaluations (89.7%), closely followed by Waley (87.0%) and Lovell (86.6%), whereas Jenner recorded significantly fewer positive evaluations (79.3%). This places Yu at the apex of the quality ranking, with a notable margin over the other translators. Beyond the quantitative findings, Yu's preeminence in translation quality stems directly from his scholarly, foreignizing translation strategy. His work, characterized by extensive paratextual elements, is consistently valued by readers as indispensable for contextual comprehension [73]. Despite occasional critiques of verbosity, these scholarly supplements bridge critical linguistic and cultural gaps while satisfying demands for authentic engagement with canonical Chinese literature [74]. This recognition is evidenced by recurring descriptive terms in reader reviews, such as "historical background" and "helpful background", alongside explicit reader affirmations including "…footnotes...necessary for complete understanding of the culture".

**3.3.3. Plot acceptance.** The 79.2% positive sentiment proportion for plot acceptance (Fig 7) indicates broad reader satisfaction with the narrative structure, reinforced by salient topic terms in Fig 8 such as "enjoy", "interest", "adventure", and "pilgrim", all underscoring affirmation of the plot's entertainment value.

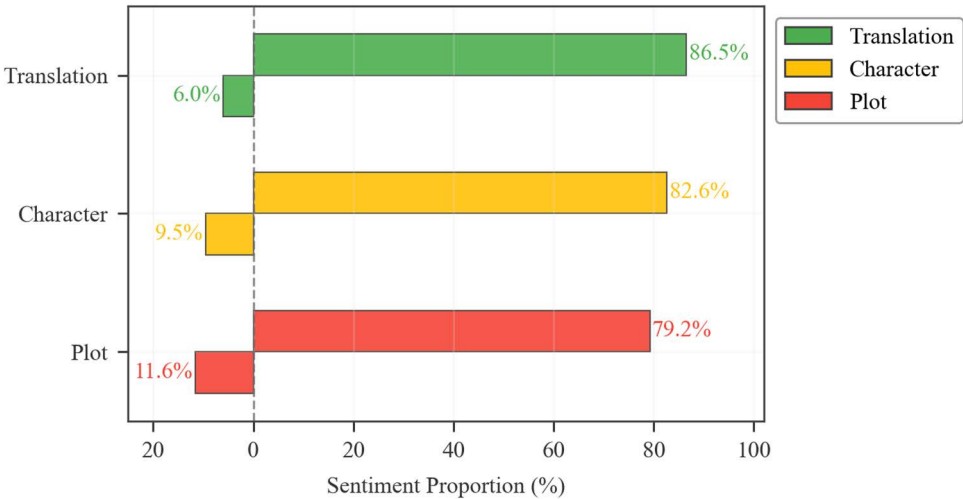

**Fig 7. Sentiment polarities in three themes. left: negative proportion; right: positive proportion.**

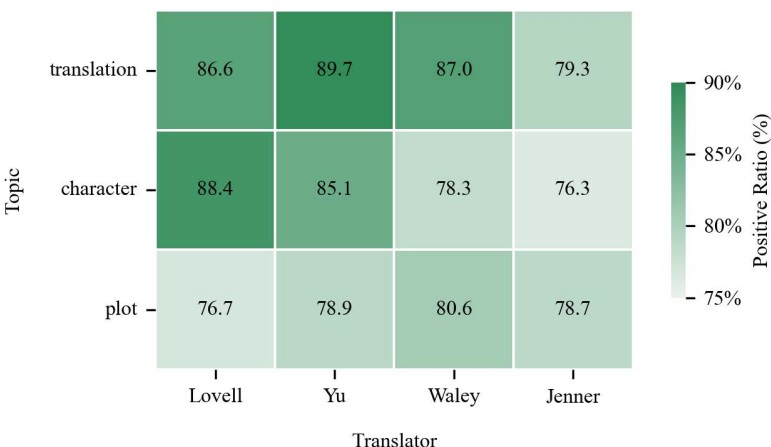

**Fig 8. Semantic network of salient cross-topic terms.** Intra-group connections are color-coded by theme, while inter-group ties appear in grey. Edge thickness scales proportionally to term co-occurrence frequency.

| Topic | Lovell | Yu | Waley | Jenner |
|---|---|---|---|---|
| translation | 86.6 | 89.7 | 87.0 | 79.3 |
| character | 88.4 | 85.1 | 78.3 | 76.3 |
| plot | 76.7 | 78.9 | 80.6 | 78.7 |

Translator

Positive Ratio (%)

**Fig 9. Distribution of positive sentiment proportions by theme and translator.**

Nevertheless, comparative analysis in Fig 7 reveals plot acceptance's negative sentiment rate (11.6%) surpasses other thematic dimensions, exceeding translation quality (6.0%) by 5.6 percentage points and character portrayal (9.5%) by 2.1 points. This significant divergence is further corroborated by the translator-level data in Fig 9, which shows that the positive sentiment proportion for plot acceptance clusters within a notably lower range (76.7% – 80.6%) across all translators, trailing behind their respective scores for translation quality and character portrayal. This consistent pattern of challenge, evident at both the macro thematic level and the individual translator level, finds its direct manifestation in reader reviews, where terms like "long" (177 times) and "repetitive" (175) epitomize criticisms of narrative pacing and structure despite overall engagement.

A significant portion of these critiques stems from two interrelated factors. First, substantial narrative excisions in translations, exemplified by Waley's abridgment of plot elements [75], may disappoint readers encountering JTTW through derivative films, TV adaptations, or video games. When such audiences seek deeper engagement with the original novel, these omissions risk failing to meet expectations for accessing its full narrative and cultural layers, provoking critiques such as: "A big downside is that it's missing so many of the best adventures…We miss out on the Monkey's crazier transformations, the beasts, gods, their magic, and interpersonal conflicts". These omissions disrupt narrative continuity and reduce character depth.

Second, the translated work's fidelity to its original episodic structure also presents significant cross-cultural challenges. The original novel employs an episodic, cyclical structure known as Zhanghui Style, a chapter-linked narrative form that prioritizes moral coherence over plot novelty [76]. In JTTW, this episodic architecture is intrinsically connected to Buddhist philosophy, where each narrative cycle symbolizes sentient beings navigating ignorance towards wisdom [77]. The "eighty-one ordeals" mirror the Buddhist theory of karmic rebirth, where repetitive practices across lifetimes are essential for liberation [78]. This structure, far from a narrative flaw, serves as a "soteriological device" [79], dismantling layers of delusion until ultimate awakening. However, it inevitably clashes with Western expectations of linear, conflict-driven plotting [80]. Even in a translation celebrated for its scholarly rigor and fidelity, such as Yu's edition, which receives the highest sentiment scores for translation quality, the positive sentiment for plot acceptance remains markedly lower (78.9%), as evidenced in Fig. 9. This clash is further reinforced by reader reviews, "…the story is long and can get repetitive at times…"; "Once the main characters are established, they have encounter after encounter with trouble which inevitably goes as follows: 1) Monster sees Tang priest; 2) Monster kidnaps Tang priest; 3) Monkey saves the day. It is in fact so formulaic…"

While these reader critiques indeed stem from a clash of cultural hermeneutic frameworks, they also, and ultimately, reflect the translational challenges inherent in structural fidelity. More specifically, these responses are symptomatic of the inherent tension translators face between domesticating the text for target-reader fluency and foreignizing it to preserve cultural specificity, a core concern in translation studies [81]. For JTTW, this tension is acute. Remaining faithful to the soteriological function of the zhanghui form risks preserving a cyclical structure that feels repetitive to readers expecting linear plots. Conversely, prioritizing readability through streamlining risks eroding the Buddhist connotations and cultural essence that define the original. This ethical obligation to preserve cultural specificity is particularly critical for classical works, where simplification can reinforce ethnocentric biases [82].

Therefore, the elevated negative sentiment toward the plot should be interpreted not as a simple measure of reader misinterpretation, but as a valuable data point. It empirically captures the enduring conflict between cultural fidelity and reader accessibility inherent in translating a culturally embedded narrative. Thus, the reception of JTTW's plot is shaped not only by cross-cultural hermeneutic gaps but, more profoundly, by the inherent and unavoidable dilemmas of the translation process itself.

**3.3.4. Character portrayal.** As shown in Fig 9, the gradient distribution of positive sentiment across translators and themes indicates that Waley's and Jenner's translations of JTTW exhibited lower positive sentiment in this regard compared to Lovell's and Yu's versions.

This discrepancy stems from their contrasting translation strategies. From reviews of Waley's translation like "…this is a superhero story. Monkey, also known as the Monkey King and Great Sage Equal to Heaven, is an immortal who has all manner of supernatural powers" and "Monkey is the original superman: a fantastical mythical tale with an east Asian flavor", it is evident that Waley's translation strategically centers Monkey as the narrative core, prioritizing his heroic exploits. At the same time, his abridgment systematically omits Buddhist terminology and secondary character arcs. Consequently, his prioritization of narrative fluency over cultural and religious fidelity results in flattened characterizations devoid of their original symbolic depth [83]. On the contrary, Lovell's translation presents more vivid characterizations through modern colloquial expressions and interactive reporting verbs [84], highlighting the dynamic and engaging portrayal of characters, as evidenced by such reviews as "Monkey is rebellious and arrogant yet heroic", and "Her translation seems particularly strong with the dialogue, being able to clearly and crisply convey the character's personalities in relation to each other, especially Monkey's habitually flippant nature".

Besides, the salient topic term "poetry" (Fig 8) indicates readers' significant engagement with literary-symbolic dimensions of verse in JTTW. This emphasis reveals divergent translation approaches between Jenner and Yu. By contrasting reviews of Yu's version—"an episodic journey simply written with intermittent vivid poetry: while visual character development isn't the focus, its presence, and the act of reading, illuminates the origins of episodic storytelling and its enduring influence",—with critiques of Jenner's translation such as "The translated poetry loses much in translation, and the actual story emerges as strikingly repetitive", the disparity in their handling of poetic elements becomes evident. Jenner prioritizes readability-over-fidelity, flattening cultural-religious components through simplified renderings and scant use of annotations [85]. Conversely, Yu's meticulous preservation of poetic and philosophical elements, coupled with scholarly paratextual apparatuses, constructs multidimensional characterizations mirroring the novel's allegorical complexity [86].

## Conclusions

This study investigates the English-language overseas online reception of *Journey to the West* by analyzing a dataset of Anglophone, platform-based reviews from Amazon and Goodreads. Grounded in this delineated scope, the research maps the complex reception landscape through a tripartite analytical framework of temporal dynamics, cross-platform sentiment patterns, and topic modeling. With a hybrid sentiment lexicon as a key methodological innovation, the study delivers three principal contributions to the study of translation reception in digital environments.

First, the research elucidates the compound drivers of reader engagement over time. While a universal increase in review counts is propelled by surging global interest in JTTW multimedia adaptations, the divergent trajectories of individual translators are primarily attributable to the specific patronage systems and accumulated cultural capital surrounding their works. Second, the analysis demonstrates that the digital platforms act as agents in reconfiguring literary valuation. The significant sentiment divergence between Amazon's e-commerce model, which prioritizes transactional utility, and Goodreads' social-literary ecosystem, centered on discursive engagement, reveals that evaluative mechanisms are deeply embedded in platform architecture. The case of Yu's translation, which exhibited a marked cross-platform variation in reception, exemplifies a Bourdieusian dynamic of cultural capital valuation reshaped by digital field infrastructures. Finally, the identification of three core reader concerns, namely, translation quality, plot acceptance, and character portrayal, pinpoints the specific loci of cross-cultural negotiation. Yu's extensively annotated translations, valued as indispensable for contextual comprehension, achieved the highest positive reception for quality. Notably, plot reception exhibited significantly heightened negativity across all translators, highlighting the enduring tension between cultural fidelity and reader accessibility as a central challenge in literary translation. Regarding character portrayal, Lovell's vivid characterizations and Yu's poetic-philosophical retention surpassed Waley's and Jenner's approaches in reader preference.

By integrating computational tools as interpretive scaffolds, this study transcends quantitative metrics to reveal cultural dynamics, demonstrating the value of such methods for digital humanities scholarship. The scope of analysis is constrained by its reliance on two major, Anglophone-centric platforms, Amazon and Goodreads. While they provide a rich

window into a significant segment of online literary reception, the findings may not fully generalize to encompass diverse digital communities and non-English-speaking contexts. Furthermore, the Amazon corpus is shaped by a platform-imposed sampling constraint. This algorithmically filtered sample may inherently over-represent highly engaged or mainstream evaluations, potentially biasing the observed time-based trends and sentiment proportions. Additionally, the domain sentiment lexicon, though tailored for this task, requires ongoing refinement to more precisely capture the nuanced and subjective ways readers evaluate literary style and narrative in translation. In light of these considerations, future research should expand the cultural and linguistic scope of this work by incorporating more diverse platforms and data sources, including non-Anglophone digital communities. To address data-specific biases such as the sampling constraints inherent to platforms like Amazon, subsequent studies could prioritize the use of existing, large-scale archival datasets. The domain lexicon, meanwhile, could be further refined through systematic, collaborative annotation involving translation scholars to deepen its adaptability to literary translations. Such advancements would enrich our grasp of how translated literature negotiates meaning across digital cultural landscapes.

## Supporting information

**S1 Table. Statistical outputs for cross-platform and cross-translator sentiment analyses.**
(XLSX)

**S1 File. Results of sentiment analysis and LDA topic modeling.**
(XLSX)

## Acknowledgments

We acknowledge the School of Foreign Languages of Wuhan Business University for hosting interdisciplinary lectures on sentiment analysis, text mining, and computational linguistics that provided methodological insights for this study.

## Author contributions

**Conceptualization:** Ningning Jia.

**Data curation:** Ningning Jia, Jing Xin, Yan Wang.

**Formal analysis:** Ningning Jia.

**Investigation:** Ningning Jia, Jing Xin.

**Methodology:** Ningning Jia, Jing Xin, Yan Wang.

**Project administration:** Ningning Jia.

**Software:** Ningning Jia.

**Visualization:** Ningning Jia.

**Writing – original draft:** Ningning Jia.

**Writing – review & editing:** Ningning Jia, Yan Wang.

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
