## [Decision Letter · Decision Letter 0]

29 Oct 2025

*Journey to the West*: Temporal dynamics, cross-platform sentiment patterns, and topic modeling: Temporal dynamics, cross-platform sentiment patterns, and topic modelingPLOS ONE

Dear Dr. Jia,

plosone@plos.org. . . . A rebuttal letter that responds to each point raised by the academic editor and reviewer(s). You should upload this letter as a separate file labeled 'Response to Reviewers'.A marked-up copy of your manuscript that highlights changes made to the original version. You should upload this as a separate file labeled 'Revised Manuscript with Track Changes'.An unmarked version of your revised paper without tracked changes. You should upload this as a separate file labeled 'Manuscript'.

We look forward to receiving your revised manuscript.

Kind regards,

Xiaoming Tian, Ph.D.

Academic Editor

PLOS ONE

Journal Requirements:

2. In your Methods section, please include additional information about your dataset and ensure that you have included a statement specifying whether the collection and analysis method complied with the terms and conditions for the source of the data.

3. Please note that PLOS One has specific guidelines on code sharing for submissions in which author-generated code underpins the findings in the manuscript. In these cases, we expect all author-generated code to be made available without restrictions upon publication of the work. Please review our guidelines at https://journals.plos.org/plosone/s/materials-and-software-sharing#loc-sharing-code and ensure that your code is shared in a way that follows best practice and facilitates reproducibility and reuse.

4. We note that Figure 1 in your submission contain copyrighted images. All PLOS content is published under the Creative Commons Attribution License (CC BY 4.0), which means that the manuscript, images, and Supporting Information files will be freely available online, and any third party is permitted to access, download, copy, distribute, and use these materials in any way, even commercially, with proper attribution. For more information, see our copyright guidelines: http://journals.plos.org/plosone/s/licenses-and-copyright.

5. We are unable to open your Supporting Information file S1_Code.py and S2_Code.py. Please kindly revise as necessary and re-upload.

6. We note that there is identifying data in the Supporting Information file “S1_Data.xlsx”. Due to the inclusion of these potentially identifying data, we have removed this file from your file inventory. Prior to sharing human research participant data, authors should consult with an ethics committee to ensure data are shared in accordance with participant consent and all applicable local laws.

-Location data

Reviewers' comments:

Reviewer's Responses to Questions

**Comments to the Author**

1. Is the manuscript technically sound, and do the data support the conclusions?

Reviewer #1: No

Reviewer #2: Yes

Reviewer #3: Partly

2. Has the statistical analysis been performed appropriately and rigorously?

Reviewer #1: Yes

Reviewer #2: Yes

Reviewer #3: Yes

3. Have the authors made all data underlying the findings in their manuscript fully available?

Reviewer #1: No

Reviewer #2: Yes

Reviewer #3: Yes

4. Is the manuscript presented in an intelligible fashion and written in standard English?

Reviewer #1: No

Reviewer #2: Yes

Reviewer #3: Yes

Reviewer #1: The conclusion section fails to provide a sufficient summary of the entire text and deviates to some extent from the original title.Some expressions are not authentic enough. For example, the abstract part could be slightly modified.

Reviewer #2: Review Report on Manuscript PONE-D-25-41113

This manuscript presents an engaging and methodologically sophisticated study on the overseas reception of Journey to the West translations. Its principal strength lies in its genuine interdisciplinary approach, integrating problems and theories in translation studies with advanced digital DH methods (e.g., hybrid sentiment analysis, LDA topic modeling). The research questions are well-defined, the analysis is robust, and the discussion provides deep interpretations that move beyond quantitative description. The construction of a domain-specific sentiment lexicon is a particularly notable methodological innovation.

While the manuscript is strong and potentially publishable a, I have several suggestions:

1. Strengthening the Literature Review and Theoretical Framing

The introduction effectively establishes the context and importance of the research. However, the review of computational methods could be updated and more tightly linked to the theoretical framework. The review of sentiment analysis and, in particular, deep learning methods (e.g., citing Tul et al. 2017) relies on literature that is becoming dated. The field of NLP has evolved rapidly with the advent of transformer-based models (e.g., BERT, GPT). Furthermore, the connection between the theoretical foundations (e.g., Reader-Response theory) and the chosen computational methods, while present, could be more explicit.

Suggestion:

Incorporate more recent reviews or studies (e.g., from 2020 onwards) that discuss the application of contextualized embeddings (BERT, etc.) and Large Language Models (LLMs) in sentiment analysis and literary text analysis. This will demonstrate a command of the state-of-the-art and provide a stronger justification for the choice of a lexicon-based approach (i.e., its interpretability and efficiency for this specific task) over more complex, data-hungry models.

Also, add a brief paragraph that more explicitly bridge the theoretical concepts with the methods. For example, clearly state how sentiment analysis operationalizes Nida’s “equivalent response,” and how topic modeling captures the “dynamic process of meaning construction” central to Reader-Response theory.

2. Deepening the Discussion and Interpretation of Findings

The discussion is generally strong but has opportunities for more nuanced analysis and concision. The explanation for cross-platform differences in Section 3.2, while valid, is somewhat repetitive in attributing the cause to both platform mechanisms and user bases, which are linked. The discussion on plot acceptance (Section 3.3.3) leans slightly towards attributing negative sentiment to reader “misinterpretation,” which could be balanced with a more translator-centered perspective. Finally, the analysis of Figure 9 (translator-by-theme sentiment) focuses almost exclusively on “character portrayal,” leaving the patterns for “translation quality” and “plot acceptance” underexplored.

Suggestion:

Refine the platform analysis: Consolidate the discussion on platform mechanisms and user bases into a more unified argument, framing the user base as a consequence of the platform’s design and primary function (e-commerce vs. social-literary).

Balance the plot acceptance discussion: While cultural hermeneutic gaps are certainly a factor, reframe the discussion to also emphasize the translational challenge this poses. The episodic structure represents a fundamental dilemma for the translator between faithfulness (foreignization) and readability (domestication). The negative sentiment is a valuable data point highlighting this enduring conflict, not just a measure of reader failure to understand.

Expand the translator-theme analysis: Provide succinct, strategy-based explanations for the observed differences in positive sentiment for each translator across all three themes. For example, briefly hypothesize why Yu's version scores highest on "translation quality" (likely due to scholarly apparatus) and why “plot acceptance” is consistently the most challenging theme for all translators.

3. Enhancing Methodological Transparency and Scope

The methods are well-described, but some choices need further justification, and the scope could be more clearly bounded. The weighted fusion scheme for the general sentiment lexicons (NRC 0.4, AFINN 0.4, VADER 0.2) is presented without empirical justification. The choice of K=3 for the LDA model is well-argued based on coherence and interpretability, but the potential value of exploring a slightly higher K (e.g., K=4) to see if a meaningful sub-topic splits from one of the three main themes is not fully dismissed. Relying on two platforms, while practical, is a limitation that should be more explicitly acknowledged upfront.

Suggestion:

Justify the lexicon weights: Either provide a reference that empirically validates this specific weighting scheme for this type of text, or briefly describe a validation step (e.g., grid search on a annotated sample) used to arrive at these values. Alternatively, clearly state that the weighting was based on a theoretical assessment of each lexicon's relative strengths.

Acknowledge LDA trade-offs: A brief sentence acknowledging that while K=3 was optimal for high-level themes, future research with different parameters might uncover more nuanced sub-themes would strengthen the methodological reasoning.

State scope limitation clearly: Explicitly mention the focus on Amazon and Goodreads as a conscious delimitation of scope in the methods section, noting they represent two major but distinct types of reception spaces (commercial vs. social-literary). This turns a potential weakness into a justified methodological choice.

Overall, this is a strong piece of interdisciplinary research that makes a valuable contribution to the fields of translation studies, digital humanities, and cultural analytics. I am confident that addressing these above points will further elevate the quality of this impressive work.

Reviewer #3: The core highlight of the paper lies in the innovation and adaptability of its methodology. Addressing issues such as insufficient domain adaptability and contextual misjudgment of general sentiment dictionaries in literary translation analysis, the researchers constructed a hybrid sentiment dictionary integrating domain-specific sentiment dictionaries with AFINN, NRC, and VADER. Through mechanisms like weighted integration, negation handling, and transition tone attenuation, it effectively resolves the semantic shift issue of words such as "literal" and "rebellious" in translation reviews. Meanwhile, combining LDA topic modeling, three core themes (translation quality, plot acceptability, and characterization) were identified under the dual verification of statistical indicators and domain interpretability. This achieves the organic unity of quantitative analysis and humanistic interpretation, breaking through the limitations of traditional translation reception studies—dominated by qualitative analysis and with limited sample sizes.

The limitations of the paper are also worthy of attention. The study covers only two English-language platforms, and the cultural representativeness of the samples needs to be expanded. Although the domain-specific sentiment dictionary has undergone double-expert annotation, it still requires larger-scale collaborative annotation for further improvement. However, the researchers have clearly proposed prospects for expanding the platform scope and optimizing dictionary adaptability in the future, leaving a clear path for subsequent studies.

Overall, taking the English translations of Journey to the West as a sample, this paper systematically depicts the complex landscape of the overseas reception of Chinese classical literature through innovations in digital humanities methods. Its triple analytical framework not only provides a reusable methodological template for translation reception studies but also deepens the interpretation of cross-cultural communication theory in the digital age by revealing the interactive relationship between platform mechanisms, translators' strategies, and readers' cognition. Against the backdrop of culture communication, such studies integrating technical tools with humanistic care undoubtedly offer important academic reference and practical enlightenment for the overseas communication and reception evaluation of classical literature.

.

Reviewer #1: No

Reviewer #2: No

Reviewer #3: No

---

## [Author Response · Author response to Decision Letter 1]

16 Dec 2025

Dear Editors and Reviewers,

Thank you for the constructive comments. We have prepared a detailed, point-by-point response to all feedback. The complete response is provided in the attached file titled "Response to Reviewers."

We appreciate your time and consideration of our revised work.

---

## [Decision Letter · Decision Letter 1]

2 Feb 2026

*Journey to the West*: Temporal dynamics, cross-platform sentiment patterns, and topic modeling: Temporal dynamics, cross-platform sentiment patterns, and topic modelingPLOS One

Dear Dr. Jia,

plosone@plos.org. . . . A letter that responds to each point raised by the academic editor and reviewer(s). You should upload this letter as a separate file labeled 'Response to Reviewers'.A marked-up copy of your manuscript that highlights changes made to the original version. You should upload this as a separate file labeled 'Revised Manuscript with Track Changes'.An unmarked version of your revised paper without tracked changes. You should upload this as a separate file labeled 'Manuscript'.

We look forward to receiving your revised manuscript.

Kind regards,

Xiaoming Tian, Ph.D.

Academic Editor

PLOS One

Journal Requirements:

Reviewers' comments:

Reviewer's Responses to Questions

**Comments to the Author**

Reviewer #2: All comments have been addressed

Reviewer #4: (No Response)

2. Is the manuscript technically sound, and do the data support the conclusions?

Reviewer #2: Yes

Reviewer #4: Partly

3. Has the statistical analysis been performed appropriately and rigorously?

Reviewer #2: Yes

Reviewer #4: Yes

4. Have the authors made all data underlying the findings in their manuscript fully available?

Reviewer #2: Yes

Reviewer #4: Yes

5. Is the manuscript presented in an intelligible fashion and written in standard English?

Reviewer #2: Yes

Reviewer #4: Yes

Reviewer #2: Reviewer Report on Revised Submission (PONE-D-25-41113R1)

1. General Evaluation

The authors have done an excellent job of addressing the concerns raised in the initial review. The revised manuscript is significantly strengthened, particularly in its theoretical framing and methodological transparency. The integration of more contemporary literature and the deepening of the discussion sections have transformed the paper from a largely descriptive quantitative study into a more nuanced interdisciplinary analysis. The authors have been very responsive to each point, providing clear justifications for their choices and making substantive changes to the text.

2. Evaluation of Specific Revision Points

Point 1: Literature Review and Theoretical Framing

• Assessment: The authors have successfully updated the literature review to include discussions of transformer-based models (BERT, GPT) and Large Language Models (LLMs). They have also provided a much-needed bridge between Reader-Response theory and their computational methods, explicitly linking sentiment analysis to Nida's "equivalent response" and topic modeling to the "dynamic process of meaning construction".

• Evaluation: This grounding provides a much stronger academic justification for the study, moving it beyond a "black box" computational exercise into a meaningful contribution to translation studies.

Point 2: Deepening Discussion and Interpretation

• Assessment: The discussion of cross-platform differences has been refined to show how platform design (e-commerce vs. social-literary) shapes user behavior. Crucially, the authors adopted a more balanced "translator-centered" perspective when discussing plot acceptance, acknowledging that reader negativity often stems from the inherent episodic structure of the text—a fundamental "foreignization" challenge for the translator—rather than simple reader misinterpretation.

• Evaluation: The expanded analysis of Figure 9 (translator-by-theme sentiment) now provides much more insight into why certain translators (like Yu) perform differently across specific themes such as "translation quality" versus "plot acceptance".

Point 3: Methodological Transparency

• Assessment: The authors have added clear justifications for their weighted fusion scheme (NRC 0.4, AFINN 0.4, VADER 0.2), citing it as a deliberate choice to balance word-level granularity with sentence-level context. They also added a thoughtful acknowledgement regarding the choice of K=3 for the LDA model, noting the trade-offs and the potential for higher K-values to reveal more granular sub-themes in future work.

• Evaluation: Explicitly stating the scope limitations regarding the use of Amazon and Goodreads further strengthens the paper by framing these platforms as deliberate choices rather than oversights.

3. Conclusion and Recommendation

The authors have addressed all major and minor concerns with rigor and clarity. The manuscript now meets the high standards of PLOS ONE for both technical soundness and humanistic interpretation.

Reviewer #4: Overall, the revised manuscript presents a clear and timely attempt to examine English-language online reception of Journey to the West translations using sentiment analysis and topic modelling. The study is potentially valuable for translation reception research and for demonstrating how computational approaches can be applied to user-generated review data. However, before the paper can be considered ready for publication, several key aspects of transparency, interpretive scope, and methodological reporting need to be strengthened.

1. First, I encourage the authors to more carefully define the scope of what is meant by “overseas reception.” While the term is repeatedly used throughout the manuscript, the empirical material analysed here consists specifically of English-language user reviews drawn from Amazon and Goodreads, which represent a particular segment of overseas reception rather than overseas reception in a broader sense. As it currently stands, the framing may lead readers to assume the findings generalise to international audiences more broadly. A clearer conceptual clarification early in the Introduction (and echoed in the Conclusion) would significantly improve the paper’s precision—for example, by explicitly describing the object of analysis as English-language overseas online reception and noting that it captures an Anglophone, platform-based subset of reception.

2. Second, the manuscript should more explicitly address the sampling constraint affecting the Amazon dataset, namely the restriction to 100 “most relevant” reviews. Although the authors mention this limitation, it deserves stronger methodological emphasis because it implies that the Amazon sample is algorithmically filtered rather than randomly or chronologically sampled, which may affect both time-based trends and sentiment proportions. At minimum, the Data section should make clear that the Amazon corpus may reflect platform ranking mechanisms, and the Limitations section should briefly discuss how this may bias observed distributions (e.g., by overrepresenting highly engaged or mainstream evaluations). This point is particularly important because the paper relies heavily on cross-platform comparison, and readers will want reassurance that platform-related sampling differences have been appropriately acknowledged.

3. Third, the credibility of the sentiment results would be substantially improved by providing greater reproducibility details for the hybrid sentiment scoring procedure. The authors propose a weighted integration of VADER, SentiWordNet, and a domain lexicon, but key implementation choices—especially the final weighting scheme (0.4/0.4/0.2), the thresholding strategy (±0.1), and the grid-search tuning procedure—are not currently described in sufficient detail for replication. I am not asking the authors to redesign the method; however, a short paragraph clarifying what optimisation criterion was used in the grid search (e.g., accuracy or macro-F1), how the tuning set was defined, and whether the overall conclusions remain stable under minor threshold or weight perturbations would greatly strengthen the methodological credibility.

4. Relatedly, the domain lexicon construction is a valuable addition, but the manuscript currently lacks information regarding annotation reliability. The authors state that two professors annotated 448 domain words on a seven-point sentiment scale, yet no inter-rater agreement statistics are reported. For an audience that may not be familiar with domain-specific lexicon construction, this omission weakens confidence in the custom lexicon’s consistency. A concise report of whether annotation was conducted independently, how disagreements were handled, and an agreement statistic (e.g., Cohen’s kappa or Krippendorff’s alpha) would be a highly feasible addition and would materially improve the rigour of this component.

5. In addition, the topic modelling results would benefit from slightly stronger evidence supporting the interpretive labelling of topics. The assigned topic names (e.g., “character portrayal,” “plot acceptance,” “translation quality”) appear plausible, but readers are not currently given enough concrete information to verify that the labels are grounded in the model outputs rather than in post hoc interpretation. A straightforward improvement would be to report the top keywords for each topic (e.g., top 10 words) and/or provide one short anonymised example excerpt per topic. This does not require expanding the analysis, but it would make the interpretive claims more transparent and more persuasive.

6. Finally, I recommend that the Discussion section adopt a somewhat more cautious tone when linking observed review trends to external cultural events (such as adaptations or major game releases). These interpretations may well be correct, but at present they read as causal explanations without direct evidence from the dataset. Since the study is observational and platform-based, the manuscript would be stronger if these points were framed explicitly as plausible associations rather than causal drivers. If feasible, the authors could also include a very light supporting check (for example, reporting whether event-related keywords appear in review texts), but even a careful wording adjustment (“may reflect,” “could be associated with”) would improve alignment between claims and evidence.

Taken together, these revisions would substantially strengthen the manuscript’s methodological transparency and ensure that the interpretive scope of the findings remains appropriately calibrated to the dataset and tools used. I believe the paper has clear potential, and addressing these points should make the contribution both more rigorous and more convincing to a broad interdisciplinary readership.

.

Reviewer #2: No

Reviewer #4: No

---

## [Author Response · Author response to Decision Letter 2]

26 Mar 2026

Dear Editors and Reviewers,

Thank you for the constructive comments. We have prepared a detailed, point-by-point

response to all feedback. The complete response is provided in the attached file entitled "Response to Reviewers."

We appreciate your time and consideration of our revised work.

---

## [Editor Report · Decision Letter 2]

30 Mar 2026

Overseas reception of English translations of *Journey to the West*: Temporal dynamics, cross-platform sentiment patterns, and topic modeling: Temporal dynamics, cross-platform sentiment patterns, and topic modeling: Temporal dynamics, cross-platform sentiment patterns, and topic modeling: Temporal dynamics, cross-platform sentiment patterns, and topic modeling

PONE-D-25-41113R2

Dear Dr. Jia,

We’re pleased to inform you that your manuscript has been judged scientifically suitable for publication and will be formally accepted for publication once it meets all outstanding technical requirements.

Kind regards,

Xiaoming Tian, Ph.D.

Academic Editor

PLOS One
---

## [Editor Report · Acceptance letter]

PONE-D-25-41113R2

PLOS One

Dear Dr. Jia,

I'm pleased to inform you that your manuscript has been deemed suitable for publication in PLOS One. Congratulations! Your manuscript is now being handed over to our production team.

Kind regards,

on behalf of

Dr. Xiaoming Tian

Academic Editor

PLOS One